# Acute Maternal Fasting or Fluid Abstention Does Not Significantly Affect the Macronutrient Composition of Human Milk: Clinical and Clinical Research Relevance

**DOI:** 10.3390/children7060060

**Published:** 2020-06-10

**Authors:** Karel Allegaert, Anne Smits

**Affiliations:** 1Department of Development and Regeneration, KU Leuven, Herestraat 49, 3000 Leuven, Belgium; anne.smits@uzleuven.be; 2Department of Pharmaceutical and Pharmacological Sciences, KU Leuven, Herestraat 49, 3000 Leuven, Belgium; 3Department of Clinical Pharmacy, Erasmus MC, Postbus 2040, 3000 CA Rotterdam, The Netherlands; 4Neonatal Intensive Care Unit, University Hospitals UZ Leuven, Herestraat 49, 3000 Leuven, Belgium

**Keywords:** lactation, physiology-based lactation models, drug exposure prediction, fasting, drug safety, newborn, infant, human milk

## Abstract

There are guidelines on lactation following maternal analgo-sedative exposure, but these do not consider the effect of maternal fasting or fluid abstention on human milk macronutrient composition. We therefore performed a structured search (PubMed) on ‘human milk composition’ and screened title, abstract and full paper on ‘fasting’ or ‘abstention’ and ‘macronutrient composition’ (lactose, protein, fat, solids, triglycerides, cholesterol). This resulted in six papers and one abstract related to religious fasting (*n* = 129 women) and observational studies in lactating women (*n* = 23, healthy volunteers, fasting). These data reflect two different ‘fasting’ patterns: an acute (18–25 h) model in 71 (healthy volunteers, Yom Kippur/Ninth of Av) women and a chronic repetitive fasting (Ramadan) model in 81 women. Changes were most related to electrolytes and were moderate and mainly in the chronic repetitive fasting model, with no clinical significant changes in macronutrients during acute fasting. We therefore conclude that neither short-term fasting nor fluid abstention (18–25 h) affect human milk macronutrient composition, so that women can be reassured when this topic was raised during consulting. Besides the nutritional relevance, this also matters, as clinical research samples—especially estimating analgo-sedative exposure by lactation—are commonly collected after maternal procedural sedation and maternal fasting. Based on these results, it is reasonable to assume stable human milk composition when such data are used in physiology-based pharmacokinetic (PBPK) models.

## 1. Introduction

Breastfeeding results in improved infant and maternal health outcomes in both the industrialized and developing world, and is recognized as an important public health issue. Consequently, women may also want to breastfeed shortly following analgo-sedation for surgery or diagnostic procedures. To further facilitate this practice, guidelines are provided by different learned societies [1,2,3]. We also recently summarized the available knowledge on lactation during or after exposure to commonly prescribed analgesics, narcotics or sedatives (opioids, intravenous and inhalational anesthetics, benzodiazepines, non-opioid analgesics, and local anesthetics) following surgery, diagnostic procedures or medical indications [4].

We proposed that the use of systemic non-opioid analgesics, local anesthetics, inhalational or intravenous anesthetics is safe when mothers are nursing. When systemic opioids are used, we recommend care providers to consider clinical monitoring of the infant for sedation. Furthermore, we suggested that duration of maternal exposure (>4 days) and the presence of maternal signs of somnolence serve as additional ‘red flags’ for adverse effects like sedation of the infant [4]. Such information is useful for lactating women and their care providers when planning analgo-sedation related to surgery or diagnostic interventions.

Following discussions with different colleagues and learned societies on anesthesia during the development of (perioperative) guidelines on lactation following sedation and clinical consulting with individual lactating women, we realized that we had only focused on drugs or compounds ‘temporarily added’ to the maternal diet, and not on the nutritional or fluid intake ‘temporarily removed’ from the maternal diet, as these women are commonly requested to abstain from enteral nutritional intake, enteral fluid intake, sometimes even water. We therefore felt that a structured search on the available evidence on the effects of maternal fasting or water abstention on the macronutrient composition of human milk is valuable to enable counseling based on the available evidence.

Along the same lines, such a structured search is also of clinical relevance as these mothers are sometimes invited to provide paired plasma and human milk samples to quantify exposure to these analgo-sedatives as part of clinical research programs. Such samples are used to explore the performance of physiology-based pharmacokinetic (PBPK) lactation models, hereby assuming that the human milk macro-nutrient composition remains stable during maternal fasting. Its valuable to confirm or reject this assumption. As recently discussed by Yeung et al., PBPK models have the ability to provide in silico estimates of drug exposure given the proper parameterization with host physiology and drug properties [5]. In order to fully exploit the utility of PBPK models in quantifying drug uptake in breastfed neonates, an accurate measure of infant feeding parameters, volume and frequency of maternal milk intake are needed [6]. Human milk composition is another of these needed parameters [5,6].

Since clinical research samples—especially for analgo-sedatives—are commonly collected after maternal procedural sedation and because these procedures are associated with maternal fasting, it is of relevance to ensure that the macronutrient composition of human milk is indeed not different from non-fasting settings so that data can be extrapolated to other populations and clinical settings and that in silico estimates of drug exposure related to lactation remain useful and reliable.

## 2. Materials and Methods

As part of a more extensive effort to explore interspecies differences in milk composition to support interspecies pharmacokinetic prediction models of drug excretion through milk with PBPK models (IMI ConcePTION project, WP3) [7,8], we conducted a search on ‘human milk composition’ within PubMed on 29 October 2019.

As an additional project of this search effort, we wanted to retrieve an answer to the question: ‘what is the impact of acute maternal fasting or fluid abstention on macronutrient (lactose, protein, fat, solids, triglycerides, cholesterol) composition of human milk’, this ‘human milk composition’ PubMed search (29 October 2019) was also screened by title, abstract and finally full paper for papers specifically related to [‘fasting’ or ‘fluid abstention’ + ‘human milk composition’] by the first author (KA). Retained papers were screened for references mentioned in these papers, and citations of these retained papers, as provided in PubMed and Google Scholar, were verified. The second author (AS) also screened the list, and cases of uncertainty were discussed between both authors.

As there is important intra- and inter-individual variability in macronutrient composition of human milk, a clinical significant change in macronutrient composition was considered to be any change beyond 1 SD compared to the non-fasting milk results, as reported in the specific, individual study retained in the analysis.

## 3. Results

The initial search on ‘human milk composition’ resulted in 3100 PubMed hits. The search strategy and screening for citations and references as described earlier, resulted in six papers and one abstract, mainly related to religious fasting (*n* = 129, Ramadan, Yom Kippur/Ninth of Av), or describing observational studies in lactating women (*n* = 23, healthy volunteers, fasting) (Table 1) [9,10,11,12,13,14,15]. In essence, there are data on two different ‘fasting’ patterns, with an acute fasting (18–25 h) model in 71 (healthy volunteers, Yom Kippur/Ninth of Av) women [9,10,11] and a more chronic, repetitive fasting (during Ramadan) model in 81 women [12,13,14,15] (Table 1, upper and lower panel respectively).

### 3.1. Acute Model

In the Yom Kippur or Ninth of Av model, an almost 25 h fast (no eating, nor drinking) has been assessed [9]. The mean changes in protein (+9%) and lactose (−6%) were smaller than the SD observed in samples collected before fasting (18.9 g/L and 8.46 µmol/L, respectively). In the Neville study (fasting 18–20 h from evening meal onwards), there were no changes in macronutrient composition in human milk [10,11].

### 3.2. Chronic, Repetitive Model

In the chronic, repetitive ‘Ramadan’ model, fasting also includes total water abstention from sunrise to sunset (daylight hours) for a month [12,13,14,15]. In the study of Prentice, this was from 5 A.M. until 7.30 P.M. [12]. However, even in the chronic Ramadan model, changes mainly related to electrolytes and were classified by the authors as moderate, while there were almost no changes in macronutrients (lactose, protein, fat, solids, triglycerides, cholesterol). When changes in macronutrients were compared to the SD values (converted from standard error of the mean when needed) reported in the milk samples, these changes were much smaller than the SD reported in the Prentice study [12]. Along the same line, Salah reported a significant decrease in lactose and protein (both −6%), but these differences in mean absolute values (−0.47 and −0.11 g/100 mL, respectively) during fasting when compared to non-fasting are ≤ or near 1 SD (0.45 and 0.22 g/100 mL, respectively) observed during non-fasting [13]. In the studies of Benner and Rakicioglu, there were no differences in macronutrients during Ramadan when compared to observations after Ramadan [14,15].

## 4. Discussion

In essence, we found data on two different ‘fasting’ patterns, with an acute model <24 h in 71 women [9,10,11], or a more chronic, repetitive fasting (during Ramadan) model in 81 women [12,13,14,15] (Table 1). In our assessment, the acute model is likely very similar to the interruption of caloric intake as part of analgo-sedation related to surgery or diagnostic interventions and strongly suggests that this is not associated with changes in macronutrient composition [1,2,3,9,10,11]. Based on the data we retrieved, the impact of chronic repetitive fasting during Ramadan on breast milk is moderate but not clinically relevant (being mainly based on morning samples, so at the start of fasting), although there remains a call for further focused research on this topic. However, this is out of the scope of the current research question with a focus on acute fasting and fluid abstention [16].

We were unable to retrieve data on human milk composition immediately following surgical interventions or medical procedures, but still feel that the pooled data in Table 1 [9,10,11,12,13,14,15] and especially the acute model data are useful for extrapolation as there is likely a major burden when conducting such studies in an immediate postoperative setting. However, this search does not exclude that the micronutrient composition of human milk is affected during either acute or chronic repetitive fasting. Furthermore, metabolic adaptation to feeding and fasting in lactating women includes increased insulin sensitivity, lower maternal insulin levels and likely also changes in the concentration of these bioactive components such as insulin in the human milk [10,11,17]. Potential changes in micronutrients or bioactive components may still have effects on infant growth and development, so this topical search cannot exclude any effect beyond the absence of changes in macronutrient composition following acute fasting [9,10,11,16].

Based on this structured, focused search, we conclude that neither acute fasting nor fluid abstention clinical significantly affect the macronutrient composition (lactose, protein, fat, solids, triglycerides, cholesterol) of human milk, so that women can be reassured on this aspect when this topic is raised during consulting. Furthermore and relevant to PBPK model development programs, this also means that—as a parameter to be used in PBPK model efforts—it is reasonable to assume a similar macronutrient composition of human milk in the immediate postoperative setting during fasting and fluid abstention [5,6,7,8].

## Figures and Tables

**Table 1 children-07-00060-t001:** Summary on study characteristics and the impact of acute (18–25 h) or chronic repetitive fasting (during Ramadan) on the macronutrients and electrolytes in human milk [9,10,11,12,13,14,15].

**Reference**	**Acute Model**	**Most Relevant Findings, Human Milk**
Zimmerman et al., 2009 [9]	48 women, nursing healthy infants (1–6 months) during a 24 h religious fasting period. Paired sampling human milk 2 days before, just after fasting, and 24–25 h later (10 mL milk before nursing).	just after vs. before: sodium (+16%); calcium (+17%); protein (+9%); phosphorus (−19%); lactose (−6%); fat unchanged. 24 h later vs. before: protein (+9%); lactose (−3%).
Neville et al., 1987 + 1993 [10,11]	23 women. Fasting after evening meal, non-caloric containing fluids allowed for 18–20 h. Repeated human milk sampling over the fasting period.	throughout fasting: milk glucose, protein, fat and lactose remained constant, despite maternal insulin and glucose decrease.
**Reference**	**Chronic, Repetitive Model**	**Most Relevant Findings, Human Milk**
Prentice et al., 1984 [12]	10 lactating women, 2 weeks before, 2nd–4th week during and 2 weeks after Ramadan; morning and evening samples. 10 non-lactating controls for maternal characteristics (morning vs. evening, lactating vs. non-lactating: higher weight loss (during the day); more dehydration; higher water turnover, likely due to higher water intake at night) in lactating women.	during Ramadan vs. before: osmolarity (+3%), sodium (+25%); lactose (−14%); potassium (−18%). during vs. after: osmolarity (+3%); sodium (+30%); lactose (−9%); potassium (−5%). during morning vs. evening, osmolarity (−3%); lactose (−12%); sodium (+55%); potassium (unchanged).
Salah et al., 2016 [13]	24 women, paired sampling during (100 mL) vs. 2 weeks after Ramadan. Morning sampling after nursing, so more a ‘chronic’ model.	during vs. after: lactose (−6%); protein (−6%); sodium (−28%); potassium (−18%); calcium (−7%); phosphorus (−14%) (fat unreported).
Bener et al., 2001 [14]	26 women, 2nd–4th week during vs. 2 weeks after Ramadan. Morning sampling after nursing, so more a ‘chronic’ model.	during vs. after: no differences in macronutrients (lactose, protein, fat, solids, triglycerides, cholesterol).
Rakicioglu et al., 2006 [15]	21 women, 2nd week during vs. 2 weeks after Ramadan. Morning sampling after nursing, so more a ‘chronic’ model.	during vs. after: no differences in macro-nutrients; potassium (−25%); dry mass (−22%). Magnesium (−12%); Zinc (−16%).

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
