# Peer review of "Acute Maternal Fasting or Fluid Abstention Does Not Significantly Affect the Macronutrient Composition of Human Milk: Clinical and Clinical Research Relevance"

_children, 2020, doi:10.3390/children7060060_

Round 1
Reviewer 1 Report
Line 24: therefore is missing an ‘e’
Line 35: maternal outcome should have an ‘s’
Line 37: consequently, women may want to breastfeed shortly after analgo-sedation for surgery or diagnostic procedures.
Line 39: sentence finishing with associations seems strange. I think a description of what kind of associations would be clearer.
Line 46: hereby deleted
Line 47: additional ‘red flags’ sentence needs finishing, i.e. additional ‘red flags’ for adverse effects on infant
Line 51: … we only had focused… should be … we had only focused…
Line 89: Table 1. Maybe group them by acute and chronic fasting. I think that would make more sense.
- Note: Maybe describe what the different fasting regimens involve a bit more i.e. what sort of fasting is the Ramadan fasting vs the other acute fasting studies.
Author Response
Line 24: therefore is missing an ‘e’ = adapted
Line 35: maternal outcome should have an ‘s’ = adapted
Line 37: consequently, women may want to breastfeed shortly after analgo-sedation for surgery or diagnostic procedures = adapted
Line 39: sentence finishing with associations seems strange. I think a description of what kind of associations would be clearer = we referred to learned societies (cfr references 1-3), adapted
Line 46: hereby deleted = adapted
Line 47: additional ‘red flags’ sentence needs finishing, i.e. additional ‘red flags’ for adverse effects on infant = adapted
Line 51: … we only had focused… should be … we had only focused… = adapted
Line 89: Table 1. Maybe group them by acute and chronic fasting. I think that would make more sense.
In the initial version of the paper, we had structured the table 1 chronologically, but we agree that there is value in a more ‘content driven’ or ‘study structure’ driven approach. Along the same line and as suggested, we have added an alinea to describe the fasting patterns specific to the studies in the results section, including Ramadan and the individual results of the studies specific to macronutrients. To further stress this, we have also adapted the title.
Note: Maybe describe what the different fasting regimens involve a bit more i.e. what sort of fasting is the Ramadan fasting vs the other acute fasting studies.
We thank the reviewer for this suggestions, and added this information in the revised version.
Reviewer 2 Report
Thank you for allowing me to review this manuscript entitled “Maternal Fasting or Fluid Abstention Does Not Significantly Affect the Macronutrient Composition of Human Milk: Clinical and Clinical Research Relevance.” This is a review of available literature on the effect of maternal fasting and fluid restriction on human milk macronutrient composition. The authors performed a PubMed search of key terms related to the subject of interest, and concluded that both acute and chronic fasting had minimal to no impact on the macronutrient content of human milk.
Overall, this is an interesting subject that has not been fully explored in order to provide sound consultation to lactating women undergoing procedures requiring brief periods of fasting and for physiology-based pharmacokinetic clinical research.
I have a few comments/questions which I outline below:
- In the methods, the authors should provide more detail regarding inclusion and exclusion criteria for the selected papers. For example, were there criteria based on study design, quality, patient sample size, etc., taken into account when determining inclusion of the paper to this review.
- Authors reported that Google Scholar was used to find articles based on citations from references found in PubMed. Did you not use Google Scholar for your initial search? Please clarify.
- For those readers unfamiliar with the fasting practices during Ramadan, Yom Kippur and the Ninth of Av, this may be a beneficial, brief detail (i.e. descriptive table) to add as the authors are hoping to extrapolate these data to preprocedural fasting.
- Table 1 – Can authors organize the table better to separate out the two fasting patterns: acute versus chronic fasting? For example, have two sections: one entitled “Acute Models” and list references 10,11, and 14; followed by another section entitled “Chronic Model” and list references 9, 12, 13, 15.
- When were samples in the Ramadan group taken in relation to pre-fast and break-fast meals specifically. A little more detail in timing of samples in relation to the fast may be helpful.
- This article mainly addressed macronutrients, but it would be of interest to note whether bioactive components such as insulin were measured, which may have implications for infant growth and body composition.
- The authors conclude that it is reasonable to assume similar macronutrient composition of human milk during fasting and fluid abstention. However, in some of the papers, they report +9% protein and -3% lactose in reference 14, -12% lactose in reference 9, and -6% in protein and lactose in reference 15. What threshold are the authors using for criteria as clinically different? A reference would be helpful.
Minor edits:
- Line 35 edit “health outcome” to “health outcomes”
- Line 36 edit “world” to “worlds”
In summary, this is a nice review of available literature on the impact both acute and chronic fasting has on the macronutrient content of human milk. More research on this subject is needed specific to preprocedural fasting, but as the authors highlight that kind of research is laborious and the data under review in the current article act as a nice surrogate.
Author Response
Overall, this is an interesting subject that has not been fully explored in order to provide sound consultation to lactating women undergoing procedures requiring brief periods of fasting and for physiology-based pharmacokinetic clinical research.
We thank the reviewer for the overall very positive assessment of this paper.
I have a few comments/questions which I outline below:
- In the methods, the authors should provide more detail regarding inclusion and exclusion criteria for the selected papers. For example, were there criteria based on study design, quality, patient sample size, etc., taken into account when determining inclusion of the paper to this review.
- Authors reported that Google Scholar was used to find articles based on citations from references found in PubMed. Did you not use Google Scholar for your initial search? Please clarify.
Response to comment 1 and 2: We have extended this paragraph, as we have used a Pubmed search on ‘human milk composition’ and screened this Pubmed search on papers related to maternal fasting or fluid abstention by title and abstract, and finally full paper. For retained papers, we have screened these papers on their references (the paper itself) and their citations (as provided in PubMed, or Google Scholar).
- For those readers unfamiliar with the fasting practices during Ramadan, Yom Kippur and the Ninth of Av, this may be a beneficial, brief detail (i.e. descriptive table) to add as the authors are hoping to extrapolate these data to preprocedural fasting.
This valuable suggestion has also been made by the first reviewer, and we have added this information to the paper.
- Table 1 – Can authors organize the table better to separate out the two fasting patterns: acute versus chronic fasting? For example, have two sections: one entitled “Acute Models” and list references 10,11, and 14; followed by another section entitled “Chronic Model” and list references 9, 12, 13, 15.
Also this request is similar to the request of the first reviewer, and we have adapted this indeed.
- When were samples in the Ramadan group taken in relation to pre-fast and break-fast meals specifically. A little more detail in timing of samples in relation to the fast may be helpful.
This was already mentioned in the table, but we have extended this information in the paper.
- This article mainly addressed macronutrients, but it would be of interest to note whether bioactive components such as insulin were measured, which may have implications for infant growth and body composition.
We agree that further exploration on mechanisms and effects is very valuable, and one of the papers has indeed explored this interaction between fasting and insulin. We have added this in the discussion section as part of a future perspective aspect.
- The authors conclude that it is reasonable to assume similar macronutrient composition of human milk during fasting and fluid abstention. However, in some of the papers, they report +9% protein and -3% lactose in reference 14, -12% lactose in reference 9, and -6% in protein and lactose in reference 15. What threshold are the authors using for criteria as clinically different? A reference would be helpful.
As we all know, there is intra- and inter-individual variability in composition of human milk, mainly driven by duration of lactation. Although arbitrary, we felt that a shift of at ≤1 SD were needed to be classified as ‘clinical different’. This has been added to the paper, and a reference on human milk composition (variability) for macronutrients has been added. This threshold has been added, while the references were the studies as retained by the search.
Minor edits:
- Line 35 edit “health outcome” to “health outcomes” = adapted
- Line 36 edit “world” to “worlds” = adapted
In summary, this is a nice review of available literature on the impact both acute and chronic fasting has on the macronutrient content of human milk. More research on this subject is needed specific to preprocedural fasting, but as the authors highlight that kind of research is laborious and the data under review in the current article act as a nice surrogate.
thank you for your kind words.